# How Would We Cycle Today If We Had the Weather of Tomorrow? An Analysis of the Impact of Climate Change on Bicycle Traffic

Anton Galich [1,*], Simon Nieland [1,*], Barbara Lenz [1] and Jan Blechschmidt [2]

[1] Institute of Transport Research, German Aerospace Center, Rudower Chaussee 7, 12489 Berlin, Germany; barbara.lenz@dlr.de

[2] INWT Statistics GmbH, Hauptstrasse 8, 10827 Berlin, Germany; j.blechschmidt@yahoo.de

* Correspondence: anton.galich@dlr.de (A.G.); simon.nieland@dlr.de (S.N.); Tel.: +49-30-6705-5507 (A.G.); +49-30-670-559-109 (S.N.)

**Abstract:** Bicycle usage is significantly affected by weather conditions. Climate change is, therefore, expected to have an impact on the volume of bicycle traffic, which is an important factor in the planning and design of bicycle infrastructures. To predict bicycle traffic in a changed climate in the city of Berlin, this paper compares a traditional statistical approach to three machine learning models. For this purpose, a cross-validation procedure is developed that evaluates model performance on the basis of prediction accuracy. XGBoost showed the best performance and is used for the prediction of bicycle counts. Our results indicate that we can expect an overall annual increase in bicycle traffic of 1–4% in the city of Berlin due to the changes in local weather conditions caused by global climate change. The biggest changes are expected to occur in the winter season with increases of 11–14% due to rising temperatures and only slight increases in precipitation.

**Keywords:** bicycle traffic; time series; weather; climate change; machine learning

## 1. Introduction

Cities around the world are facing challenges due to climate change and are increasingly required to formulate and implement adaptation strategies for changed climate conditions [1]. In transportation planning, this is particularly important for trips that are greatly affected by weather conditions, such as cycling trips. Many cities promote cycling as a mode of transport as it does not emit carbon or any other harmful air pollutants and contributes to a healthier lifestyle by increasing the amount of daily physical activity [2]. Therefore, planning future bicycle infrastructure will benefit from understanding the variations in bicycle usage brought about by the characteristics and the extent of climate change.

However, the scholarly literature on the impacts of climate change on bicycle usage is rather thin. Indeed, it is well understood how cultural, infrastructural, economic, and sociodemographic factors affect bicycle usage [3–9]. Furthermore, many studies looked into the impact of weather conditions on cycling rates and found significant effects of temperature [10–14], precipitation [9,15–17], wind speed [18–20], snow [21,22], and humidity [6]. Rising temperatures generally tend to increase bicycle traffic, while adverse weather conditions such as precipitation, snow, and high humidity usually have a negative impact on bicycle usage. Ref. [23] provides a more comprehensive review of the weather conditions whose impact on bicycle traffic was analysed.

In contrast, only two studies explicitly address the potential impact of climate change on bicycle usage with different approaches. Ref. [24] compared people's mobility behaviour in years that weather-wise resemble the conditions that are expected to be brought

about by climate change to that in years which resemble average seasonal weather conditions at the moment in order to highlight the differences. Ref. [25] trained a negative binomial model on bicycle count station data and current weather conditions and applied it to predict future cycling rates on the basis of future weather conditions generated by climate models.

Against this background, this study aims at enriching the scholarly literature by using machine learning prediction algorithms and data from regional climate models to analyse the potential impact of climate change on bicycle usage. For this purpose, we focus on the changes in air temperature, precipitation, and wind speed as predicted by regional climate models, while keeping constant all other relevant factors such as infrastructure, economic development, etc. In this way, it is analysed how we would cycle today if we had the weather of tomorrow.

Hence, we follow the conceptual framework put forward by [25] but attempt to improve its accuracy by using machine learning models, which are known for their excellent prediction accuracy [26,27]. We first train four different time series models (seasonal autoregressive integrated moving average with exogenous factors (SARIMAX), Facebook Prophet (Prophet), XGBoost, long short-term memory neural network (LSTM)) on local weather and bicycle count data for the city of Berlin. Hence, we compare the traditional statistical approach SARIMAX to three are machine learning models suitable for analysing time series data. In a second step, we select the model with the highest prediction accuracy (XGBoost) and apply it to predict bicycle traffic based on the altered weather conditions that we expect in the future due to climate change.

The study is organised as follows. Section 2 presents the database and highlights necessary pre-processing and data fusion. Section 3 introduces the modelling approach. Section 4 presents the results, which are further discussed in Section 5, before conclusions for practitioners are drawn in Section 6.

## 2. Materials

Bicycle counts, weather, and climate data constitute the primary database for our study, which is described in more detail in the following two subsections. First, we explain how the data from the bicycle count stations in Berlin were prepared and aggregated to the daily level for further analysis. Thereafter, we provide a brief overview of the sources of weather and climate data used in this paper. In particular, we explain how the projections of future weather conditions for regional climate models are used in combination with the local weather conditions measured in Berlin in 2017, 2018, and 2019 to generate three scenarios of future weather conditions that we expect to be brought about by climate change.

### 2.1. Bicycle Count Data

Bicycles constitute an important means of transport in Berlin and account for 15% of the modal split of all trips conducted on a typical day [28]. For a more precise picture of its bicycle traffic, the city of Berlin started to install automatic bicycle count stations in 2012 [29]. Altogether, 26 bicycle count devices were installed throughout the city by 2020. Each device automatically detects the bicycles passing by and reports the aggregated number of cyclists per hour.

After data preparation and the exclusion of count stations with too many missing values (see Appendix A for more details), 17 bicycle count devices remained. Figure 1 shows the locations of the remaining bicycle count devices.

The count stations cover the whole area of the city including central, northern, eastern, southern, and western parts. Furthermore, most of the count stations are located in areas in which large shares of the population of Berlin live (see also Figure A2 in the Appendix A). Therefore, we assume that they represent the entire volume of bicycle traffic in Berlin quite well so that daily, weekly, monthly, seasonal, or yearly variations in the actual volume of bicycle traffic are also present in our data. Our rather conservative

approach to data preparation gave rise to a data set with consistent patterns across the different count stations.

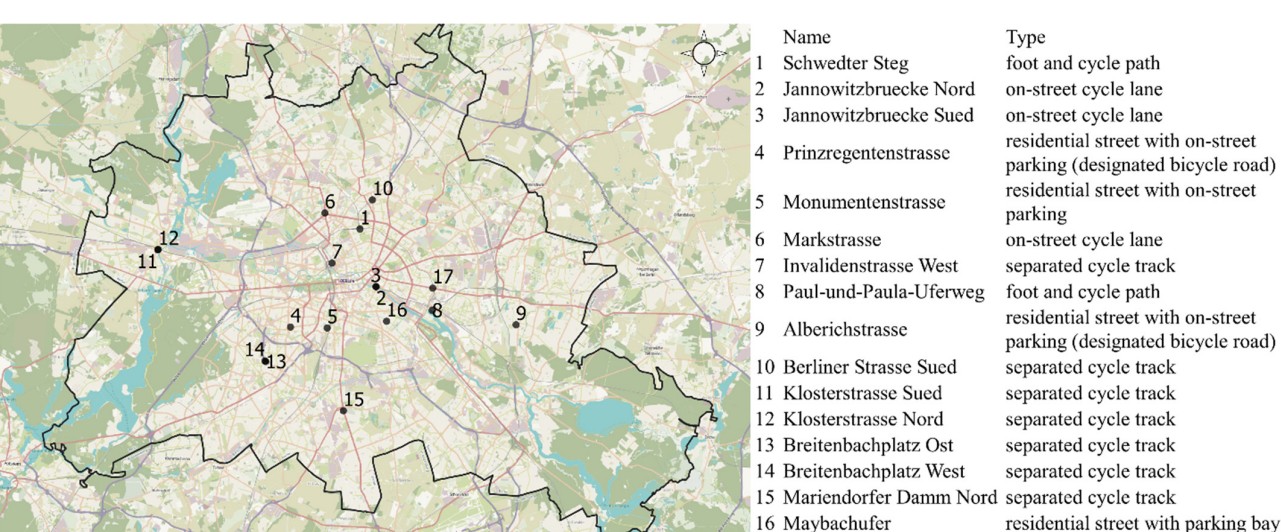

| | Name | Type |
|---|---|---|
| 1 | Schwedter Steg | foot and cycle path |
| 2 | Jannowitzbruecke Nord | on-street cycle lane |
| 3 | Jannowitzbruecke Sued | on-street cycle lane |
| 4 | Prinzregentenstrasse | residential street with on-street parking (designated bicycle road) |
| 5 | Monumentenstrasse | residential street with on-street parking |
| 6 | Markstrasse | on-street cycle lane |
| 7 | Invalidenstrasse West | separated cycle track |
| 8 | Paul-und-Paula-Uferweg | foot and cycle path |
| 9 | Alberichstrasse | residential street with on-street parking (designated bicycle road) |
| 10 | Berliner Strasse Sued | separated cycle track |
| 11 | Klosterstrasse Sued | separated cycle track |
| 12 | Klosterstrasse Nord | separated cycle track |
| 13 | Breitenbachplatz Ost | separated cycle track |
| 14 | Breitenbachplatz West | separated cycle track |
| 15 | Mariendorfer Damm Nord | separated cycle track |
| 16 | Maybachufer | residential street with parking bay |
| 17 | Frankfurter Allee West | separated cycle track |

**Figure 1.** Locations of bicycle count stations in Berlin and corresponding infrastructure type.

After data preparation, various statistical tests were performed to check the stationarity of the time series and to analyse whether the bicycle counts at the different count stations follow similar seasonal patterns and react to weather conditions in a similar manner. In particular, we looked at the mean and variance of the different count stations, computed correlation matrices and correlation coefficients, and conducted the Augmented Dickey–Fuller Test and the Kwiatkowski–Philipps–Schmidt–Shin Test. The details of this procedure are described in Appendix A.

All of these analyses have shown that our data can be regarded as stationary. In general, stationarity is an important concept in time series analysis and denotes that the statistical properties of a time series such as the mean and the variance do not change over time. Stationarity also implies that there is no trend in the time series data such as a linear growth trend, for example, which leads to increasing values in the data from year to year. In our case, stationarity implies among other things, that the volume of bicycle traffic measured at each count station reacts in a very similar way to changes in weather conditions over the entire period of time covered by our data.

The bicycle counts from these remaining devices were aggregated to a daily level for two reasons. First, we are interested in predicting the impact of climate change on the total volume of bicycle traffic in Berlin and not on the number of bicycle counts at individual stations. Second, the climate change data are generally provided on that level. There are procedures to narrow the data down to the hourly level but these would increase the uncertainty considerably.

Our data show a mean of 34,452 bicycle rides per day with a standard deviation of 479. The maximum number of bicycle rides was recorded on 21 June 2017 at 71,367, while the minimum was detected on 8 January 2017 at 2864. The total number of bicycle counts per year varied from 11.7 million in 2017 to 13.1 million in 2018 and 13.0 million in 2019. Figure A1 in the appendix provides a graphical illustration of our data with plots of the daily bicycle counts at each station as well as the aggregated number of counts over all stations.

## 2.2. Using Outputs from Regional Climate Models to Generate Future Weather Scenarios

In order to predict the impact of climate change on the volume of bicycle traffic, we generate changed local weather conditions for the year 2050 according to the three different Representative Concentration Pathways of the IPCC: RCP2.6, RCP4.5, and RCP8.5. RCPs are trajectories of greenhouse gas concentration in the atmosphere describing different climatic futures adapted by the IPCC.

These emission scenarios for climate change should not be seen as forecasts or predictions of future climate conditions but rather as expert judgements of plausible future emissions based on socioeconomical, environmental, and technological developments incorporated in integrated assessment models. Since the future evolution of anthropogenic factors cannot be known in advance, the possible effects are presented in different scenarios describing several possible emission pathways [30].

RCP2.6 (meaning the radiative forcing is 2.6 Watts per square metre ($W/m^2$) in 2100) shows a peak in greenhouse emission concentration around 2040 and then declines, RCP4.6 stabilises at ~4.5 ($W/m^2$) after 2100, and RCP8.5 increases constantly even after 2100. Therefore, RCP8.5 can be regarded as a "business as usual" scenario in which no climate protection actions are realised, even after 2100. RCP2.6 represents a "moderate" emission scenario and RCP2.6 can be seen as a "climate protection" scenario. For more detailed information on RCPs and their societal and environmental implications, see [30].

This study uses outputs of a regional climate model (domain: EUR-11, driving model name: MPI-ESM-LR, realisation: r1i1p1, frequency: day) developed and calculated by the EURO-CORDEX initiative, which aims at downscaling global climate projections to a regional scale (12 km ground resolution) for the European continent [31–33]. In fact, the EURO-CORDEX initiative ran many different simulations with various regional climate models downscaling the output of 10 different global climate models. In order to decrease uncertainty and to avoid biases due to the features of a particular model, this study uses the output of one of those models whose simulations of maximum air temperature, sum of precipitation, and mean wind speed most often fell into the range of median values of the outputs of the different models for the three RCPs. To produce realistic climatic scenarios, the characteristics of their weather conditions need to be consistent with the actual weather in Berlin. Realistic refers to yearly distributions of weather conditions in terms of their mean, variance, and range as well as the number of extreme weather events such as hot days (≥30 °C) or days with heavy rain (≥10 mm). To do this, we adapt the output of regional climate models that predict the expected climatic changes to local weather conditions that were measured in Berlin in 2017, 2018, and 2019.

Therefore, we carry out the following data preparation steps: First, we link daily measurements from a local weather station to the bicycle counts to generate the training basis for the forecast model (see Section 3.2). Second, we calculate the absolute changes in weather conditions between historical and forecast periods from a regional climate model to extract the expected impact of climate change. Third, we adapt these expected changes to the weather conditions measured in Berlin in 2017, 2018, and 2019 in order to generate realistic local weather conditions for 2050. The details of this procedure are described in Appendix B.

The results of the data processing are synthetic distributions of the weather variables for each day of the year, reflecting the expected changes in weather from the future climate scenarios of RCP2.6, RCP4.5, and RCP8.5. These synthetic distributions will be fed into our time series model to predict the volume of bicycle traffic in 2020 under the changed weather conditions that we expect climate change to bring about. Figure 2 illustrates the results of our adaptation procedure for the RCP8.5 scenario and outlines the synthetic future values generated in relation to the weather station's average daily maximum air temperature, sum of precipitation, and mean wind speed values in 2017, 2018, and 2019.

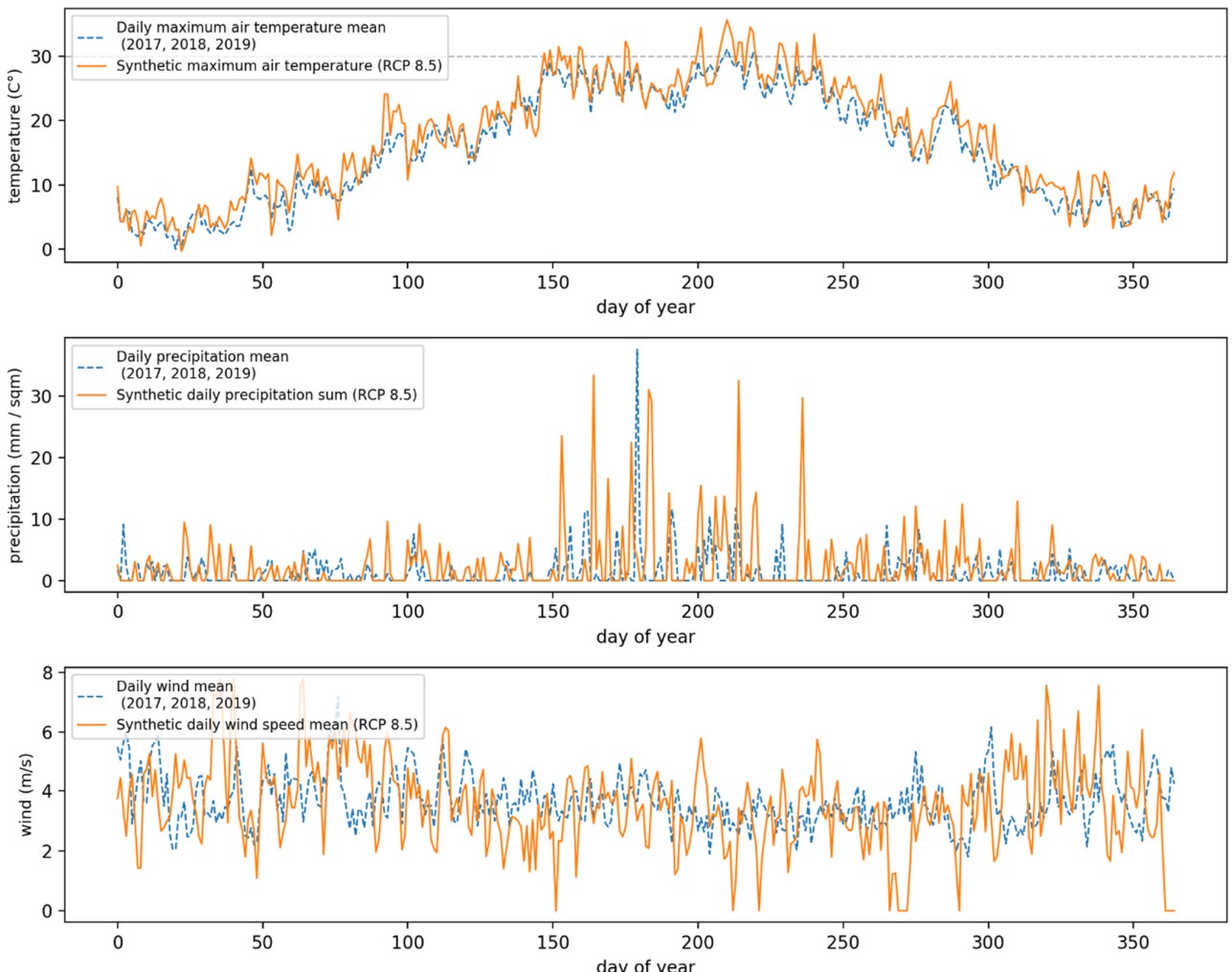

**Figure 2.** Yearly distributions of the synthetically generated future weather conditions and averaged weather station data from 2017, 2018, and 2019. The dashed blue line shows the reference value of the respective weather variable (top: maximum air temperature averaged over 2017, 2018, and 2019; centre: absolute precipitation averaged over 2017, 2018, and 2019; bottom: absolute wind speed averaged over 2017, 2018, and 2019). The orange line illustrates the yearly distributions of the synthetically generated weather variables based on the data from the RCP8.5 scenario. The RCP2.6 and RCP4.5 scenarios are excluded from the figure for illustrative purposes.

As can be seen, our adaptation procedure resulted in realistic yearly distributions of the three weather variables: maximum air temperature, sum of precipitation, and mean wind speed (see also the results section for a more quantitative comparison). Realistic in this case means two things. First, the data meet the expected changes to be brought about by the different climate change scenarios in the number of extreme weather events as defined by the German Weather Service [34,35], such as hot days (≥30 °C) or days with heavy rain (≥10 mm) (see [33]). Second, the mean, the variation, and the range of the daily values of the three weather variables also reflect the expected changes to be brought about by the different climate change scenarios [33]. Hence, our adaptation procedure resulted in three different yearly distributions for 2020 that realistically portray the mean changes in local weather conditions in Berlin that can be expected until 2050 on the basis of RCP2.6, RCP4.5, and RCP8.5.

### 3. Methods

In this section, we explain our methodological framework in more detail. For this purpose, we first illustrate the general research design of this study. Subsequently, the four time series models that were chosen for comparison are described before we illustrate the cross-validation procedure developed for model selection. Finally, it is outlined what independent variables were used (see Table 1) and how the different models were tuned on the basis of the cross-validation procedure.

*3.1. Research Design*

Figure 3 illustrates the research design of this study including model selection, data preparation, and prediction.

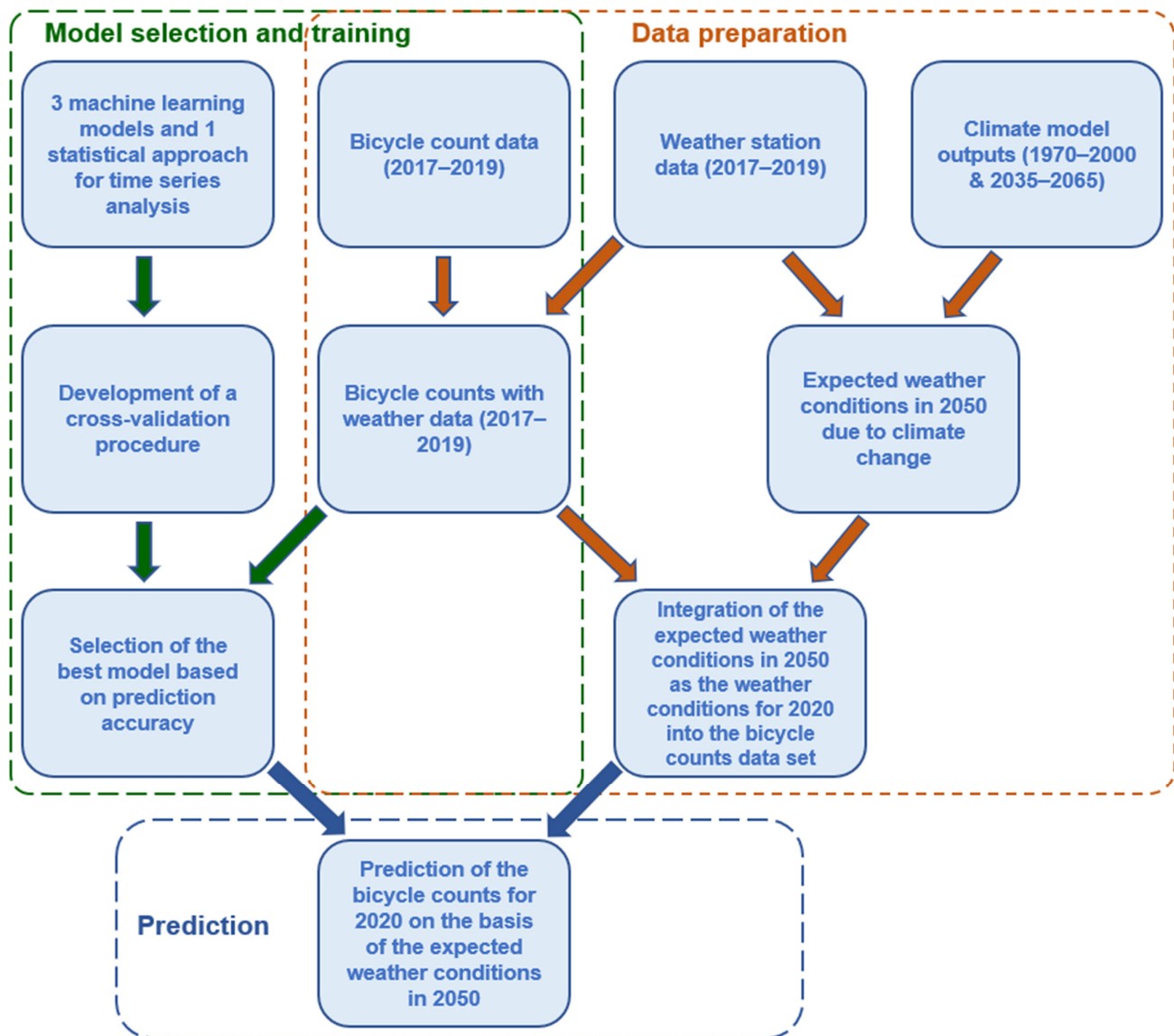

**Figure 3.** Research design. Green arrows indicate processes that refer to model selection and training, while red arrows denote data preparation steps and blue arrows refer to prediction processes.

We chose to compare three machine learning models to one traditional statistical approach for time series analysis which can be seen as a sort of benchmark. For this

comparison, a cross-validation procedure was developed based on the bicycle count data of 2017–2019 which were enriched with the measured weather conditions for the same period. The cross-validation procedure was used to tune the four models and to select the best model on the basis of prediction accuracy.

The trained algorithm is then used to predict bicycle counts for a future year using synthetically generated weather conditions. As we have measured weather data and bicycle counts for 2017, 2018, and 2019, we predict the bicycle traffic volume of 2020 under the changed climatic conditions of 2050 considering the three climate change scenarios RCP2.6, RCP4.5, and RCP8.5 in order to generate a consistent time series. This means that we use the different models to predict bicycle volume in 2020 in combination with weather data for 2050.

*3.2. Forecast Models*

In the following, we describe the four models that we chose to compare in this study. All four methods were selected due to their ability to model time series data. As the objective is to find the best method for the prediction of bicycle counts, prediction accuracy was used as the central benchmark in the cross-validation procedure. This suggests the use of machine learning models instead of traditional statistical approaches, as the former are directly optimised on increasing the prediction accuracy, while the latter are optimised on improving model fit [36]. Thus, three machine learning models and one traditional statistical approach are compared with regard to their prediction accuracy in this paper.

Seasonal autoregressive integrated moving average with exogenous factors (SARIMAX) constitutes the classic statistical approach for modelling time series data with seasonality [37]. SARIMAX models are parametric approaches based on the concepts of autoregression and moving averages [37]. As such, they allow a relatively straightforward interpretation of the relationships between the independent and the dependent variables and constitute a popular tool for analyses of time series data which aim at clarifying these relationships [6].

Prophet is a generalised additive model developed by Facebook for time series analyses [38]. It is a non-parametric approach in which the dependent variable is estimated by an addition of functions of the trend in the data, the seasonality, and the holidays as well as an error term [38]. While the trend function uses either a logistic or a linear growth model to capture the trend in the data accurately, the seasonal function relies on Fourier series to model periodic seasonality [38]. The holiday function incorporates the changes induced by specific, reoccurring events such as public holiday by assigning each holiday a parameter corresponding to its impact on the dependent variable [38]. These features make prophet models more flexible in terms of detecting complex relationship patterns in the analysis of time series data than SARIMAX approaches, while preserving some degree of the interpretability.

XGBoost constitutes a regression tree-based ensemble algorithm with a gradient boosting framework [39]. Hence, XGBoost is used for fitting multiple regression trees, each based on a random subsample of the data, which are pooled into a so-called ensemble. The boosting principle denotes that each tree is built on previous trees with particular emphasis on the mis-modelled values so that the overall prediction performance improves gradually. Thus, each tree constitutes a separate function and the final prediction output of the dependent variable is the sum of these functions [39]. These mechanisms increase the predictive power of XGBoost, yet they also decrease its interpretability. While approaches to measure the importance of the independent variables in predicting the dependent one have been developed, these interpretations rely solely on the data used and do not make any assumptions on general distributions of the variables beyond the data set [40].

Long short-term memory is an artificial recurrent neural network architecture which is well suited for predictions based on time series data due to its reliance on sequences rather than single values [41,42]. As all neural networks in general, lstm is a highly flexible

model able to adapt to diverse and complex patterns in different kinds of data ranging from tabular data (as in our case) to audio or visual data [41,42]. Yet, this flexibility comes at the price of decreased interpretability. Indeed, in spite of increasing efforts to open up the black box of neural networks [43], it still remains difficult to reveal the detected patterns in the data analysed in an understandable manner.

### 3.3. Cross-Validation

For selection of the best model, we developed a cross-validation procedure based on the concepts of simulated historical forecasts and rolling windows. This means that first, we had to specify a cut-off date. Then, we performed the first training on the data from the first date the data provide up to this cut-off date, and we made predictions for a specified time period of the remaining data after the cut-off date, called the forecast horizon. During consecutive training, the cut-off date was shifted forward in time by half the number of days of the forecast horizon and we repeated training based on the increased training data set before making predictions for the new forecast horizon, which extended into the future from the new cut-off date.

The development of this cross-validation procedure included the testing of many different specifications. It was, for instance, also tried to shift the cut-off date forward day by day, by a week, a month, a quarter of the year, etc. In fact, often the results in terms of the prediction accuracy did not differ that much between different cross-validation approaches. However, the cross-validation procedure presented in this paper is the one that generally led to the best results, even if not by large margins.

To provide an example of the cross-validation procedure developed: If 1 July 2017 was chosen as the first cut-off date and the forecast horizon was specified by 30 days, then the first training was performed on data from 1 January 2017 to 1 July 2017 and predictions were made for the first 30 consecutive days after 1 July 2017. For the next training, the cut-off date was shifted by half the number of days of the forecast horizon to 16 July 2017, the algorithm was trained on data from 1 January 2017 to this new cut-off date, and predictions were made for the first 30 consecutive days after 16 July 2017. This process was repeated until the shift of the forecast horizon extended beyond the last date of our data set (31 December 2019).

The final step in our cross-validation procedure was to calculate the mean absolute percentage error. This was realised by first calculating the mean absolute percentage error for each specific day of the predictions in consecutive order, then taking the mean of the mean absolute percentage error for the same days of the different prediction rounds, and finally taking the mean of the mean absolute percentage error for each day of the forecast horizon. In the example above, this means first calculating the mean absolute percentage error for the first day of the forecast horizon for the different predictions, then taking the mean of these calculations, repeating the same procedure for each other day of the forecast horizon, and finally taking the mean of the mean absolute percentage errors for each day of the forecast horizon.

### 3.4. Model Tuning

Based on our cross-validation procedure, we tuned the four different methods with the ambition of minimising the mean absolute percentage error for a forecast horizon of 365 days because our objective was to use the best model for the prediction year of 2020. All models were initially tested with the same set of features, including the numerical variables maximal air temperature, sum of precipitation, and mean wind speed as well as the categorical variables day of the week, month, and year, and the dummy variables public holiday and school holiday. Table 1 provides an overview of the independent and dependent variables:

**Table 1.** Independent and dependent variables.

| Name | Data Type | Unit | Temporal Resolution |
|---|---|---|---|
| **Independent variables** | | | |
| Maximal air temperature | Numeric | ° Celsius | Daily (maximum) |
| Sum of precipitation | Numeric | Millimetres | Daily (sum) |
| Mean wind speed | Numeric | Metre per second | Daily (mean) |
| Day of the week | Categorical | - | Daily |
| Month | Categorical | - | Monthly |
| Year | Categorical | - | Yearly |
| Public holiday | Dummy | - | Daily |
| School holiday | Dummy | - | Daily |
| **Dependent variable** | | | |
| Bicycle counts | Numeric | - | Daily |

As single days constitute our temporal level of analysis, each feature comprises 1095 observations that together cover the three years 2017, 2018, and 2019. As there is no linear trend in the bicycle counts in Berlin (see Section 2.1), we do not have to apply differencing transformations to our data. Consequently, it also means that our predictions for the year 2020 will have no linear trend that would predict a rise or decline in the bicycle counts, irrespective of the weather.

We applied the auto.arima function from the forecast package in R to evaluate the SARIMAX method. As the auto.arima function automatically detects the best combinations of $p$ (autoregression order), $d$ (difference order), and $q$ (moving average order) values as well as the seasonal components $P$ (seasonal autoregression order), $D$ (seasonal difference order), $Q$ (seasonal moving average order), and $m$ (the number of time steps for a single seasonal period) based on the Bayesian Information Criterion (BIC), we did not perform any further manual parameter tuning for the SARIMAX method. The best model had the configuration of SARIMAX(3, 1, 1) (0, 1, 0 (365)) and a BIC of 15,826.

Prophet was developed with the specific intention of providing a tool that works quite well out of the box. Thus, using its implementation in the prophet package in R, we allowed the model to automatically detect the yearly, weekly, and daily seasonalities in our data set (with the default Fourier order of 10). We also tested the model by manually adding monthly seasonalities with a Fourier order of 12 and chose the additive instead of the multiplicative seasonal effect because we did not observe any changes in the strength of the seasonal patterns over the three years in our data set. We also chose the linear growth model because in general, it still seems more appropriate for the development of bicycle counts than the logistic growth model, although we did not detect a linear trend in the three years of our data.

For XGBoost, implementation in the library named likewise in Python was used. We chose the "dart" booster and linear regression as the objective. Since XGBoost was not developed to automatically detect specific patterns in time series data, categorical variables for the day of the week, month, and year were added to the training data set as well as the first lag of our bicycle count data.

For training of the LSTM, based on the keras and tensorflow packages in R, variables such as day of the week, month, and year were transformed into binary subvectors using one-hot-encoding. Together with the numerical features (air temperature, precipitation, and wind speed), these one-hot-encoded variables were then scaled based on the minimum and maximum of the training data to a range of −1 to 1. In a manual trial and error approach, different configurations with one and two hidden layers; a batch size of one, five, and 73; SGD, Adam, and Adamax as optimisers; dropout rates of 0.05, 0.1, 0.2, 0.3, and 0.5; varying numbers of units in the first and the second hidden layer were tested with 50, 100, 500, 1000, and 2000 epochs. All algorithms were tested with and without weather information.

## 4. Results

This section first describes the results of our model comparison and selection. Thereafter, the prediction results of the XGBoost model are illustrated for the bicycle traffic in 2020 under the synthetically generated weather conditions.

### 4.1. Model Performance, Comparison and Selection

We used our cross-validation procedure to perform an exhaustive grid search for Prophet and XGBoost to find the optimal configuration of hyperparameters. For Prophet, this resulted in 5 for seasonality.prior.scale, 0.05 for changepoint.prior.scale, and 20 for holidays.prior.scale. The optimal values for XGBoost were 0.01 for eta, 0 for gamma, 15 for max_depth, 5 for min_child_weight, 0.5 for subsample, and 1 for colsample_bytree. The manual trial and error approach applied for the LSTM resulted in two hidden layers with 100 and 50 units, dropout rates of 0.2, a batch size of 5, the Adam optimiser, the mean absolute error as loss function, the hyperbolic tangent activation function, and 50 epochs yielding the lowest mean absolute percentage error.

Figure 4 shows the results of our cross-validation procedure for the best model specifications of all four methods for six different forecast horizons.

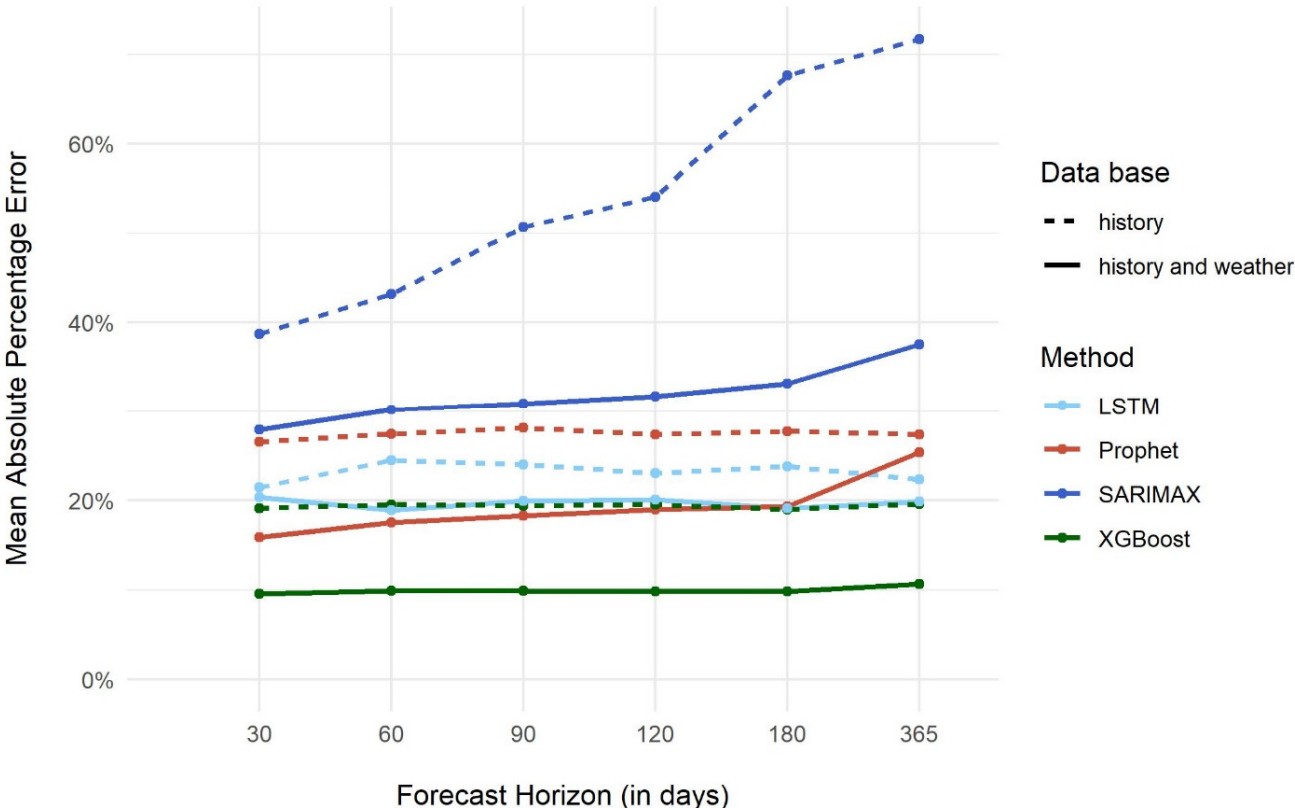

**Figure 4.** Cross-validation results for different forecast horizons. The colours represent the different methods compared. The dashed lines show the mean absolute percentage error when only the bicycle counts were used for prediction. The solid line illustrates the mean absolute percentage error for the predictions based on bicycle counts and weather variables.

The prediction accuracy of all four methods improves considerably if not only the bicycle count data are used for training but also the weather station measurements on daily maximum air temperature, sum of precipitation, and mean wind speed. This illustrates once more the importance of including weather conditions when modelling bicycle traffic. In particular, the parametric SARIMAX approach improves considerably when the weather conditions are included as further independent variables in the model.

Furthermore, it can be seen that the prediction error increases for all methods if the forecast horizon is extended to 365 days.

Our XGBoost configuration, trained on the bicycle counts and the weather station measurements, clearly yields the best prediction accuracy for all six forecast horizons. Its mean absolute percentage error is 11% for the forecast horizon of 365 days, while the errors for LSTM, Prophet, and SARIMAX are 20%, 25%, and 38% respectively. XGBoost was therefore chosen for prediction of the future bicycle counts based on the synthetically generated future weather conditions derived from global and regional climate science models.

Unfortunately, the relatively high prediction accuracy of machine learning approaches comes at the disadvantage of a more difficult interpretation of the results. Indeed, it is rather difficult to illustrate why one model performs better than the other. We assume, however, that the structure of our data in terms of size and heterogeneity is not complex enough to fully exploit the advantages of neural network approaches such as LSTM. These constitute generally the most flexible machine learning methods being able to adapt relatively well to complex non-linear relationships in the data. Yet, our data only comprised a very limited number of variables and cases by machine learning standards.

In contrast, XGBoost is based on regression-trees which in our case appears to better capture the underlying relationships in our data. This might be due to the fact that the relationship between temperature and bicycle traffic is of a rather linear nature. However, the relatively bad performance of the SARIMAX approach indicates that there are revenant non-linear relationships in our data that are better captured by non-parametric machine learnings methods than by traditional parametric statistical approaches.

### 4.2. Predictions

Figure 5 illustrates the actual bicycle count data of 2017, 2018, and 2019 and the predicted bicycle counts per day for the year 2020 on the basis of the weather conditions of the three different RCPs in 2050.

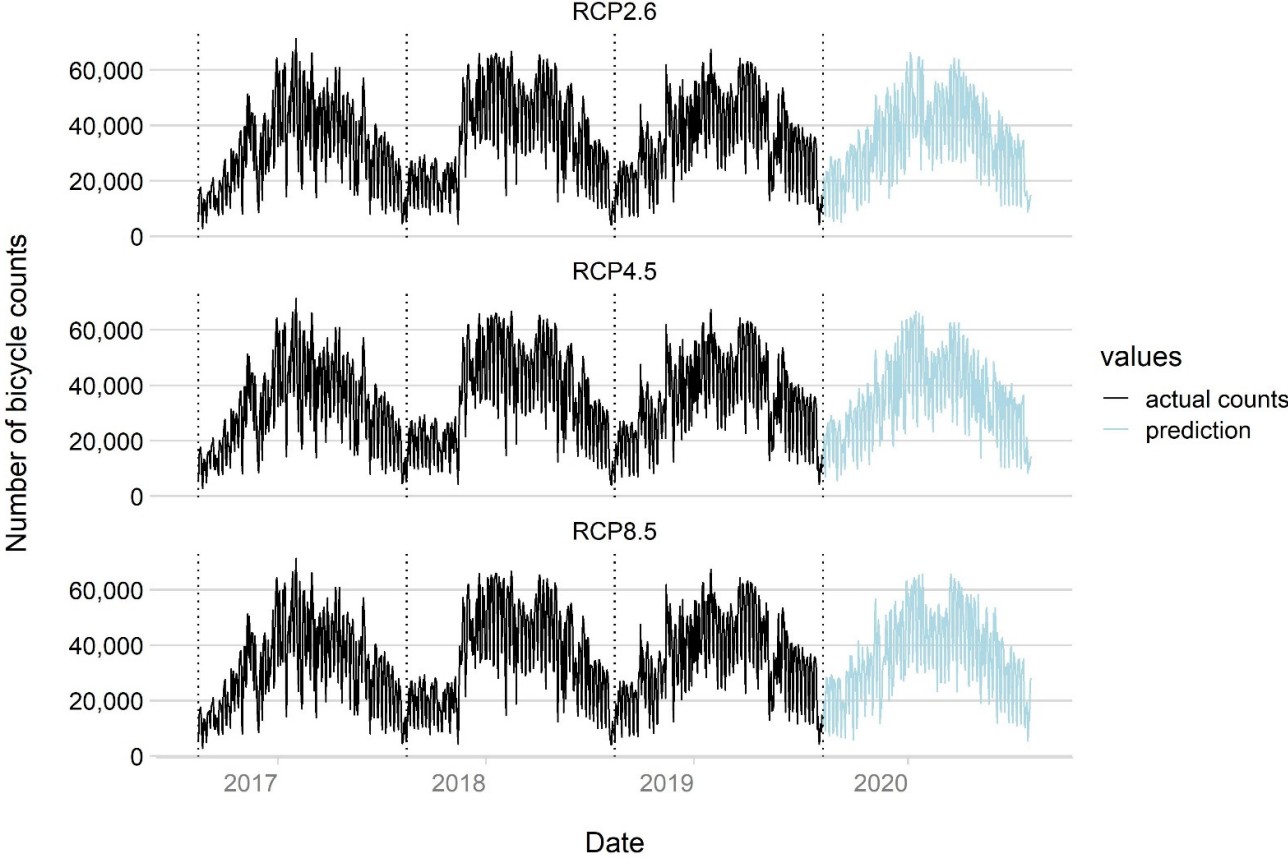

**Figure 5.** Actual daily bicycle counts and future predictions. The black line illustrates the actual bicycle counts detected in 2017–2019. Therefore, these values are the same ones for all three RCPs. The light blue line shows the prediction of the bicycle counts which different among the RCPs due to different weather conditions.

Our model seems to capture the general seasonal patterns quite well and to some extent also accounts for the expectable daily variations. Table 2 summarises the synthetically generated future weather conditions for 2020 and the predicted bicycle traffic for the three representative concentration pathways compared to the weather station measurements and the actual bicycle count data of 2017, 2018, and 2019.

**Table 2.** Weather station measurements, synthetically generated future weather conditions, actual bicycle counts, and prediction results.

| Season | 2017 | 2018 | 2019 | 2020 (RCP2.6) | 2020 (RCP4.5) | 2020 (RCP8.5) |
|---|---|---|---|---|---|---|
| **Maximal daily air temperature in °Celsius (mean)** | | | | | | |
| Spring | 15.35 | 16.38 | 15.21 | 15.94 | 16.34 | 16.98 |
| Summer | 24.18 | 27.19 | 27.30 | 26.43 | 27.65 | 27.92 |
| Autumn | 14.22 | 16.11 | 15.17 | 16.18 | 15.92 | 17.19 |
| Winter | 4.19 | 5.17 | 6.65 | 5.74 | 5.97 | 6.52 |
| **Daily precipitation in mm (sum)** | | | | | | |
| Spring | 89.8 | 103.5 | 91 | 120.92 | 116.36 | 129.18 |
| Summer | 400.4 | 97 | 198.6 | 393.33 | 349.76 | 401.68 |
| Autumn | 191.4 | 55.7 | 140.1 | 194.54 | 246.66 | 208.18 |
| Winter | 114.8 | 119.7 | 94.8 | 137.18 | 113.84 | 125.99 |
| Total | 796.4 | 375.9 | 524.5 | 845.86 | 825.61 | 865.01 |
| **Daily wind speed in m/s (mean)** | | | | | | |

| | | | | | | |
|---|---|---|---|---|---|---|
| Spring | 3.85 | 3.88 | 4.23 | 3.95 | 3.90 | 3.92 |
| Summer | 3.35 | 3.47 | 3.26 | 3.23 | 3.17 | 3.24 |
| Autumn | 3.62 | 3.39 | 3.22 | 3.41 | 3.32 | 3.35 |
| Winter | 3.91 | 3.86 | 3.87 | 3.76 | 3.71 | 3.70 |
| **Specific weather events (number of days)** | | | | | | |
| Hot days * (≥30 °C) | 8 | 31 | 26 | 16 | 24 | 26 |
| Dry days * | 171 | 240 | 207 | 213 | 227 | 209 |
| Days with heavy rain * | 22 | 7 | 13 | 20 | 17 | 19 |
| Days with moderate or strong winds * | 40 | 34 | 37 | 37 | 37 | 34 |
| **Bicycle counts (sum)** | | | | | | |
| Spring | 3,189,554 | 3,369,275 | 3,301,122 | 3,215,209 | 3,317,420 | 3,340,132 |
| Summer | 4,014,512 | 4,487,628 | 4,386,066 | 4,281,880 | 4,317,930 | 4,377,109 |
| Autumn | 3,004,748 | 3,487,274 | 3,268,441 | 3,285,980 | 3,222,224 | 3,396,897 |
| Winter | 1,450,916 | 1,755,762 | 2,009,325 | 1,931,819 | 1,974,888 | 1,982,382 |
| Total | 11,659,730 | 13,099,939 | 12,964,954 | 12,714,888 | 12,832,461 | 13,096,522 |

* Hot days refers to days with a maximum air temperature of at least 30 °C. Dry days are days with less than 0.1 mm precipitation. Days with heavy rain refers to days with a precipitation of at least 10 mm. Days with moderate or strong winds refers to days with a mean wind speed of at least 5.5 m/s (Beaufort 4).

In a nutshell, the results indicate that climate change will contribute to an overall rise in bicycle traffic in Berlin if all other relevant factors are held constant. More specifically, we expect an increase in bicycle traffic compared to the mean total of 12,574,874 for the three years 2017, 2018, and 2019 for all representative concentration pathways (RCP2.6: +140,013; RCP4.5: +257,587; RCP8.5: +521,647). This corresponds to an overall expected relative increase of 1.1% for RCP2.6, 2.1% for RCP4.5, and 4.1% for RCP8.5. This general increase, however, varies significantly between the seasons and is slowed down partly by the expected increase in precipitation.

We expect the highest rise in bicycle traffic in the winter season. Compared to the mean bicycle traffic for the winters of 2017, 2018, and 2019, our model predicts an increase of 11.1% in RCP2.6, 13.6% in RCP4.5, and 14.0% in RCP8.5. The main reason for this is the expected increase in air temperature in winter. Its positive impact on bicycle traffic outweighs the negative impact of the increase in precipitation that is also expected. In addition, comparison of the future predictions with the actual bicycle counts highlights the very high bicycle traffic in the exceptionally warm and dry winter months of 2019. Only the winter conditions in RCP8.5 lead to similar bicycle traffic, while the bicycle traffic predicted for a typical winter season in RCP2.6 and RCP4.5 remains below the level of 2019.

This large increase in bicycle traffic expected during winter outweighs all the changes in bicycle traffic predicted for the other seasons together. In a typical spring season in RCP2.6, we expect a decrease in bicycle traffic by 2.2% due to relatively high precipitation. In RCP4.5, the average maximum air temperature rises a bit, while precipitation is a bit lower than in RCP2.6, so an increase of 0.9% in bicycle traffic is predicted. In a typical spring season in RCP8.5, bicycle traffic is predicted to increase by 1.6% as the effects of higher air temperatures outweigh the impact of the increasing precipitation.

In fact, the relative changes in air temperature and precipitation due to climate change are the most important factors affecting the increase or decrease in bicycle traffic predicted for the different seasons. In summer and autumn in RCP2.6 and RCP4.5, the average maximum daily air temperature is expected to rise a bit but the expected relative increase in precipitation is even higher with the result that the predicted overall changes in bicycle traffic remain at a low level. For a typical summer season, our model predicts a

decrease of 0.3% in RCP2.6 and an increase of 0.5% in RCP4.5, while in autumn, bicycle traffic is predicted to increase by 1.0% in RCP2.6 and to decrease by 1.0% in RCP4.5. In contrast, in RCP8.5, bicycle traffic is predicted to increase by 1.9% in summer and by 4.4% in autumn, reflecting, above all, the higher increases in the average maximum daily air temperature in comparison to RCP2.6 and RCP4.5.

## 5. Discussion

Our results show that climate change will lead to an overall increase in annual bicycle traffic in Berlin of between 1% and 4%. During winter, in particular, bicycle traffic might increase by 11–14% due to higher air temperatures and only a relatively low increase in precipitation. Although increases in air temperature are also expected in the other seasons, their positive effect on bicycle traffic is offset by relatively high increases in precipitation, which can even lead to a decrease in predicted bicycle traffic for some seasons, considering the representative concentration pathways 2.6 and 4.5.

This overall positive effect of climate change on bicycle usage in Berlin corresponds to the findings of [24] for the region of Randstad in the Netherlands. In addition, [24] also found that climate change might lead to a higher bicycle usage in winter due to milder temperatures and only slightly more precipitation. However, the results of [24] for the summer season show a decrease in bicycle traffic due to more intense precipitation expected to be brought about by climate change. The findings of our study also differ a bit from those of Wadud [25], produced for bicycle traffic in London. First, our expected annual increase with 1–4% is larger than his of 0.5% [25]. Second, he predicts the largest seasonal increase of 2.5% for the season of summer, while we expect the largest increases to occur in future winters [25].

The differences in our results and the other two studies could be based on the different historical weather data for Berlin, London, and Randstad, on different changes in the local weather expected to be brought about by climate change due to geographical differences, or on different reactions of cyclists to weather conditions in the three locations. Thus, the comparison of the results should be treated with care and rather illustrates the importance of further research on the impact of weather conditions on mobility behaviour in different climate zones.

However, given the fact that nearly all studies worldwide observe similar effects of weather patterns on cycling rates [23], it can be assumed that cities with similar climatic conditions (humid continental climate with dry winters and warm summers [44]) as Berlin and similar expected changes due to climate change can also expect similar impacts on cycling rates. This counts especially for European cities with a continental climate such as Prague, Warsaw, Vienna, Bratislava, Budapest, Kiev, etc.

Furthermore, it should be kept in mind that the annual changes in bicycle traffic might also be affected by population growth. Based on the data of the Office for Statistics for the region of Berlin and Brandenburg [45], we calculated annual growth rates of 1.08%, 0.87%, and 0.68% for the years 2017–2019. These are modest growth rates but other cities might have a more dynamic population growth, in which case, this factor should definitely be addressed in the analysis and also in the comparison with the results of this study.

Finally, the reliability of the results of this study also depends on the outputs of the climate change models. As already illustrated, regional and global climate models rely on various assumptions about ecological, economic, social, and technological developments and thus naturally come with a lot of uncertainties. To account for this uncertainty, various regional climate models based on the output of different global climate models simulating three different scenarios were run and compared with each other. This constituted the state-of-the-art procedure in climate science at the moment when this study was conducted.

## 6. Conclusions

In this paper, we introduce machine learning methods for the prediction of future bicycle traffic based on bicycle count, weather station, and regional climate model data. Our results have shown a higher prediction accuracy of machine learnings methods in comparison to a traditional statistical approach for time series modelling. This should be considered by future studies that are more interested in predicting accurate results based on well-known relationships between dependent and independent variables than on exploring the nature of these relationships.

Furthermore, in contrast to model fit measures such as the Akaike information criterion, which is often used for model selection in traditional statistical approaches, but which cannot be directly calculated for non-parametric approaches, we developed a cross-validation procedure with the mean absolute percentage error as the central benchmark for model selection. This increases the comparability of the performance of our methodological framework to future studies as the mean absolute percentage error can easily be calculated for both parametric and non-parametric approaches.

Our results show that bicycle traffic in Berlin will most likely increase due to the effects of climate change, if all other factors remain constant. City planners should consider these findings since they need to prepare infrastructures that are suitable for changes in demand and allow for increased requirements for more sustainable mobility. In particular, the expected increase in cycling in the darker winter months contributes to a higher traffic load on bicycle lanes throughout the year, providing an additional argument for the further extension of street lighting and bicycle infrastructure in general. This might also increase public and political acceptance of the need to redistribute public space for bicycle usage.

In addition to the specific results, our research also highlights the importance of including weather conditions in any analysis of mobility behaviour in general. The prediction accuracy of all four methods compared improved considerably if not only bicycle count data were used for training but also the weather information regarding maximum daily air temperature, sum of precipitation, and mean wind speed. This not only illustrates the importance of weather conditions for bicycle traffic but also paves the way for further research which might investigate what other modes of transport benefit or suffer from increases or decreases in bicycle traffic due to climate change, and to what extent different groups of people, in terms of age, gender, etc., adjust their mobility behaviour to weather conditions.

However, the inclusion of weather variables has also illustrated the limitations of our approach. Our findings on the potential impact of climate change on bicycle traffic in Berlin are hardly comparable to the results of similar studies on other locations due to different present weather conditions and differing impacts of climate change. Therefore, more studies in different geographic and climatic regions are needed to better understand the impacts that climate change might have on mobility behaviour in different parts of the world.

**Author Contributions:** A.G.: Conceptualisation, Methodology, Software, Validation, Data Curation, Writing—Original Draft Preparation. S.N.: Conceptualisation, Methodology, Software, Data Curation, Writing—Original Draft Preparation. B.L.: Conceptualisation, Writing—Review and Editing. J.B.: Methodology, Formal Analysis, Writing—Review and Editing. All authors have read and agreed to the published version of the manuscript.

**Funding:** This study was produced within the scope of the Helmholtz Association's mobility subproject for the Climate Initiative.

**Data Availability Statement:** The bicycle count data used in this paper are publicly accessible via the homepage of the company Eco-counter (https://www.eco-public.com/ParcPublic/?id=4728 (accessed on 21 February 2021)). The weather station data can be retrieved in the open data portal of the German Weather Service (https://www.dwd.de/DE/klimaumwelt/cdc/cdc_node.html (accessed on 20 November 2020)). The data outputs of the regional climate models of the German Climate

Service Centre (GERICS) used for this study can generally be accessed via the Euro-Cordex project (https://euro-cordex.net/060378/index.php.en (accessed on 25 November 2020)). However, at the moment of the submission of this paper, the concrete data used have not yet been published.

**Acknowledgments:** We would like to express our thanks to Thomas Remke from the German Climate Service Centre (GERICS) who not only provided the data for the representative concentration pathways for us but also gave us advice on how to adjust these data to our weather station measurements for the region of Berlin. We would also like to thank Michael Hardinghaus for carefully reading the manuscript and giving valuable advice on all parts of this paper.

**Conflicts of Interest:** The authors declare that they have no known competing financial interests or personal relationships that could have appeared to influence the work reported in this paper.

## Appendix A

As the count devices were installed at different points in time and as some of them were occasionally out of operation or showed unreasonable results, some data preparation was needed to produce a consistent data set for our analysis. As our objective was to predict the volume of bicycle traffic for each day of an entire year, we decided to train our algorithm on data available for entire years. This left us with data for the years 2017, 2018, and 2019, as around half of the stations went into operation in 2016. Four count devices were excluded from further analysis because they showed missing values over several days, weeks, or months in a row.

Missing values at the remaining stations were imputed by taking the mean of the previous and following hour. If missing values occurred at 11 pm on 31 December 2019, the value of the previous hour was used. Days with zero bicycle counts were regarded as extremely unlikely under regular conditions and thus taken as an indicator for irregularities. Three stations showed consecutive days with zero bicycle counts over several months. Two other stations showed extremely low or extremely high values over several months. In both cases, these time periods were deemed too long for data imputation and thus, all five stations were excluded from further analysis.

Figure A1 illustrates the sum of the bicycle counts per day per stations for the remaining 17 devices after data preparation.

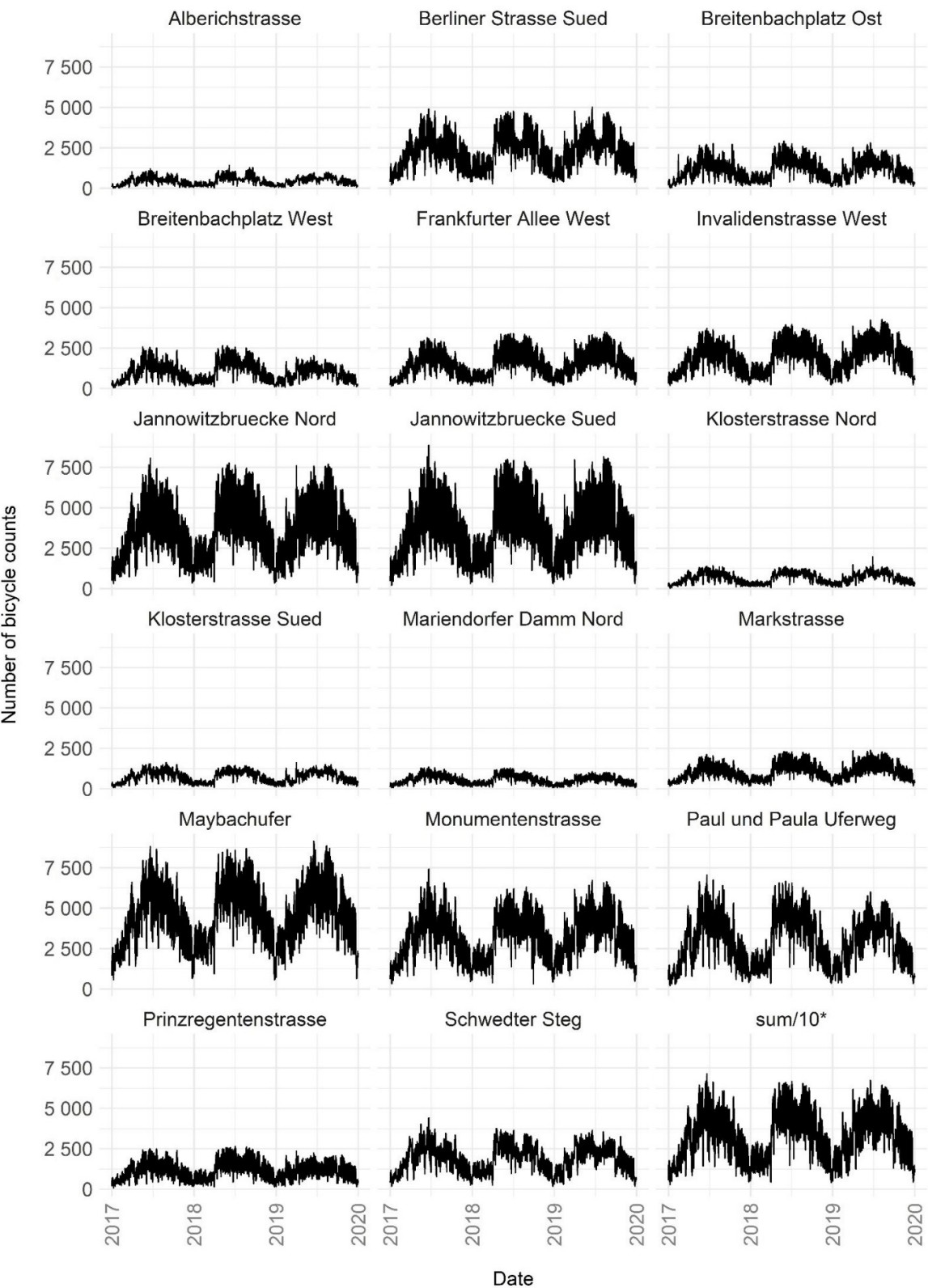

**Figure A1.** Bicycle counts for each count station per day from 2017 to 2020. * The aggregated number of all stations on the bottom right of the figure was divided by ten in order to fit on the same scale as the counts from the individual stations.

We first analysed whether the bicycle counts at the different count stations follow similar seasonal patterns and react to weather conditions in a similar manner. For this purpose, various statistical tests were conducted to check the stationarity of the time series of the different count stations. As can be seen in Figure A1, each station shows the same general seasonal trend with peaks in summer and the lowest number of bicycle counts during the winter months and has a relatively constant mean and variance per year. In addition, the Augmented Dickey–Fuller Tests and Kwiatkowski–Philipps–Schmidt–Shin Tests performed had *p*-values smaller than 0.05, allowing us to reject the null hypotheses of a unit root or trend-stationarity being present. Therefore, the time series of the different count stations can be regarded as stationary. This allows us to investigate the correlation between the bicycle counts per day at the different stations directly without the need of detrending the data first.

In fact, the lowest value in the correlation matrix of all count stations is the Pearson correlation coefficient of 0.87 between the count stations of Alberichstrasse and Frankfurter Allee West. For the combinations of the large majority of all count stations, the correlation coefficient has a value higher than 0.9. Consequently, the seasonal patterns of the individual count stations are also visible in the bicycle counts aggregated over all stations, shown in the bottom right corner of Figure 2.

The count station data are not just highly correlated between the individual stations but also the counts of the different stations show a very similar correlation to the weather variables taken from the weather station Berlin-Tempelhof of the German Weather Service. Pearson's correlation coefficient between the maximum daily temperature and the daily bicycle count of each station falls into the range of 0.60 to 0.82. Moreover, the correlation between the bicycle counts of the individual stations per day and the sum of precipitation per day and the mean wind speed per day fall into the rather narrow ranges of −0.08 to −0.12 and −0.21 to −0.26. This shows that the volume of bicycle traffic measured at each count station reacts in a very similar way to changes in weather conditions.

We also inspected whether the locations of the remaining count stations after data preparation correspond to the population distribution in Berlin:

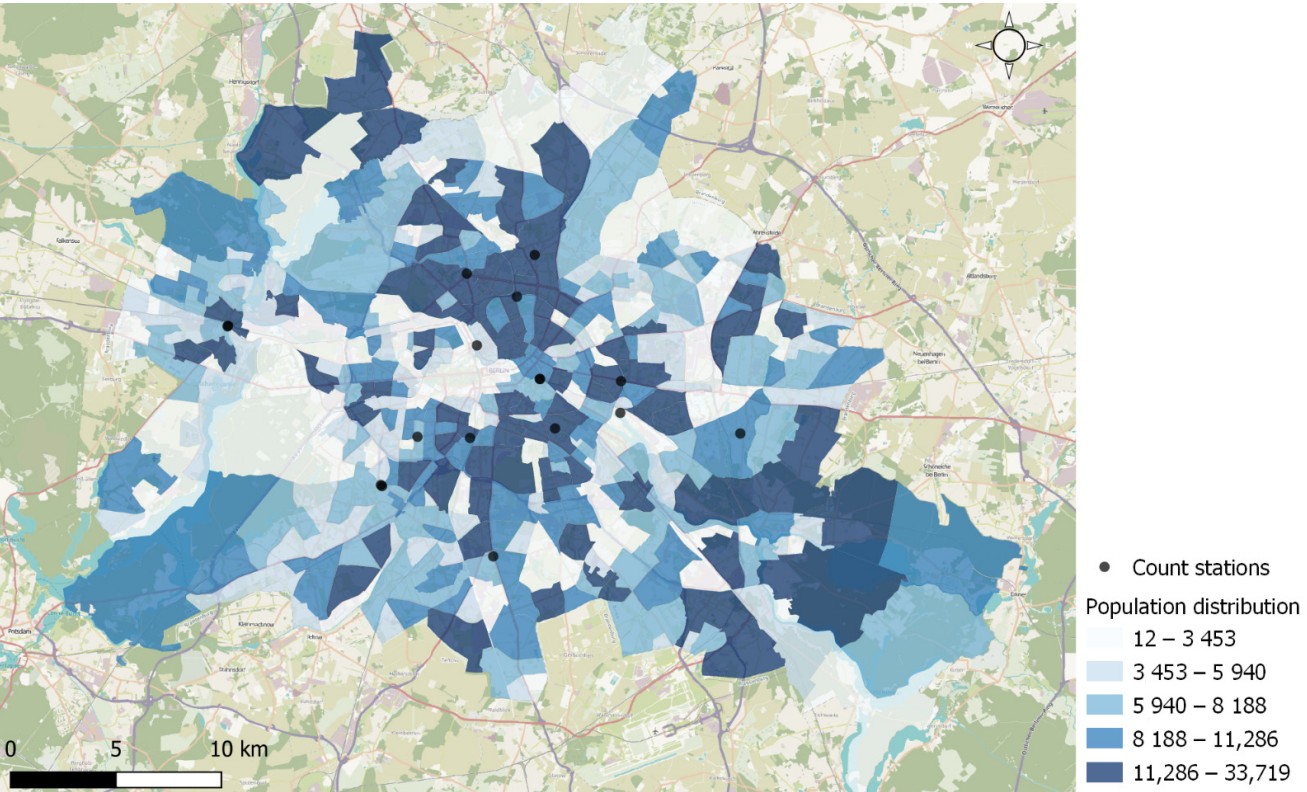

**Figure A2.** Count station locations and population distribution.

As can be seen in Figure A2, most of the count stations are located in areas where also large shares of the population reside. Only some areas with relatively many inhabitants in the north and in the east and southeast are not covered or not covered well.

### Appendix B

Two types of weather and climate data were used in this paper: daily measurements from a nearby weather station (further called weather station measurements) and weather projections of a regional climate model for the time periods 1970–2000 (further called historical data) and 2035–2065 (further called forecast data). All data sources include the three weather variables of maximum air temperature, sum of precipitation, and mean wind speed for each day of the year (see Figure A2).

The weather station measurements are taken from the Berlin-Tempelhof station for 2017, 2018, and 2019. The station is operated by the German Weather Service (DWD) and provides data on an hourly and daily basis. For this study, we use the daily measurements for each day of the three years considered. These data are linked to the daily bicycle count data by their date to provide the basis for training the time series models.

To make reasonable assumptions about future changes in the weather, this study applies outputs from the regional climate model (REMO) [46] (domain: EUR-11, driving model name: MPI-ESM-LR, realisation: r1i1p1, frequency: day) for the historical data and the forecast data, considering the three different Representative Concentration Pathways (RCP): RCP2.6, RCP4.5, and RCP8.5.

The data of regional climate models are available in approximately 12.5 × 12.5 km ground resolution. In order to reflect the local uncertainties of the climate models, a mask of 6 × 6 pixels (appr. 75 × 75 km) is placed around Berlin. Finally, we use the mean of the resulting data.

Figure A3 illustrates the need to adapt the outputs of the regional climate model to the local weather conditions. This is performed by comparing the yearly distribution of

the historical and forecast results from the regional climate model with the distribution of the measured values from the weather station in Berlin in 2018. Generally, the daily maximum air temperature from the regional climate model shows higher values than the measured weather data in 2018 since they illustrate the highest values of a 30-year interval instead of one single year. However, the absolute changes from historical to forecast data appear realistic, as they lead to an overall annual increase of 1.6 °C in the average maximum daily air temperature, which is consistent with the output of recent findings for Germany (Brasseur et al., 2017).

Therefore, in the case of air temperature, we can simply add the expected absolute changes from the climate models to the averaged weather station data to create realistic synthetic future temperature conditions for 2050 (see Figure A3). In contrast to air temperature, however, wind speed and precipitation are less continuous over time and rainy days and storms are mostly discrete events. This leads to an unwanted effect when averaging the historical and forecast data: the values are smoothed. Just adding the differences means there would be no cases of heavy rain and no cases without rain in the data set, leading to a bias in the variance in respect of single-year data. Hence, a more sophisticated procedure is required to adapt the expected changes in precipitation and wind speed from the climate models to the weather station data.

For this reason, the synthetic future precipitation values are generated using a variance adjustment procedure to harmonise measured and forecast yearly distributions. It can be assumed, however, that there are different effects on the variation of precipitation based on the time of year. Findings from the regional climate simulations in Germany (Pfeifer et al., 2015), for instance, suggest that summer precipitation is expected to decrease, while winter precipitation is expected to increase. Therefore, the measured, historical, and forecast data were first split into the meteorological seasons of winter (1 December to 28 February), spring (1 March to 30 May), summer (1 June to 30 August) and autumn (1 September to 30 November).

For each season, the absolute differences between the precipitation sums of the historical and forecast data are calculated. Here, we use the season with the highest variance because extreme events are expected to increase in the future and we want to adapt to the most "extreme" season we have. The expected absolute differences in precipitation are then added to the sum of the corresponding season in the measured weather station data to generate an expected future precipitation sum for the respective RCP. Then, we calculate the proportion of precipitation for each day in relation to the sum of precipitation for the respective season and multiply the result by the expected future precipitation sum. The outcome of this process is the expected daily precipitation with the distribution of the respective measured season.

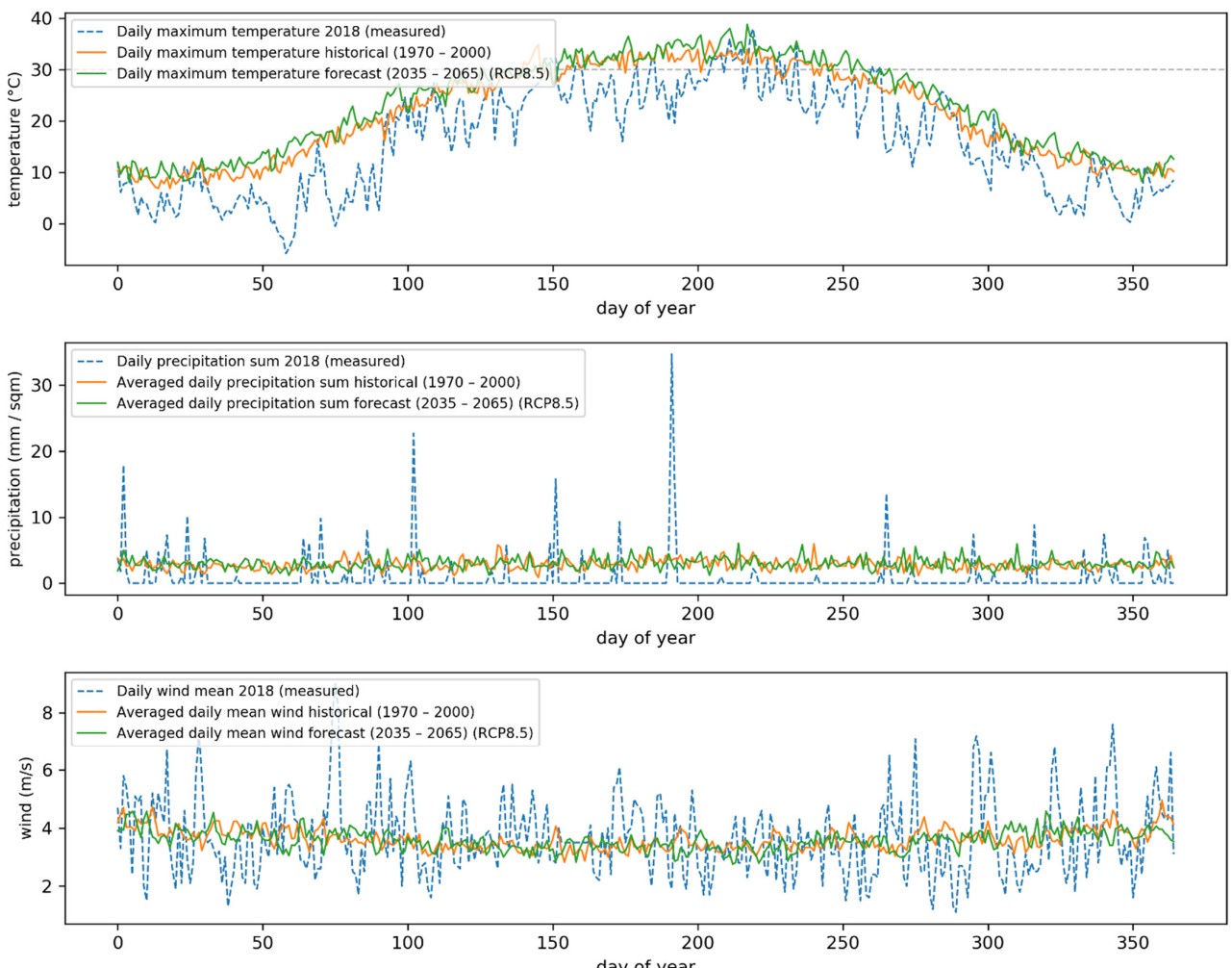

**Figure A3.** Yearly distribution of the historical (orange) and forecast (green) weather variables based on the regional climate model and the measured weather variables for 2018 (blue dashed) from a local weather station. Forecast values are based on RCP.

Next, the variance of this synthetically generated future precipitation data is adjusted to meet the probability distribution of the year with the highest variance in the respective season (winter and spring of 2018 and summer and autumn of 2017) in order to achieve a realistic (but highly variable) distribution of precipitation. The variance adjustment is realised by performing the following steps. First, the synthetically generated precipitation values are centred by subtracting the yearly mean of each value. Second, the centred values are multiplied by the square root of the variance of the measured data divided by the variance of the forecast data. Third, the data are brought out of centre again by adding the yearly mean to each value. Since the results include negative values, the data are split into positive and negative values. The latter are iteratively eliminated by setting negative values to zero and subsequently adding the proportion of the sum of all negative values in relation to the sum of positive values to each positive value as a fourth step. This step is performed until all negative values are eliminated.

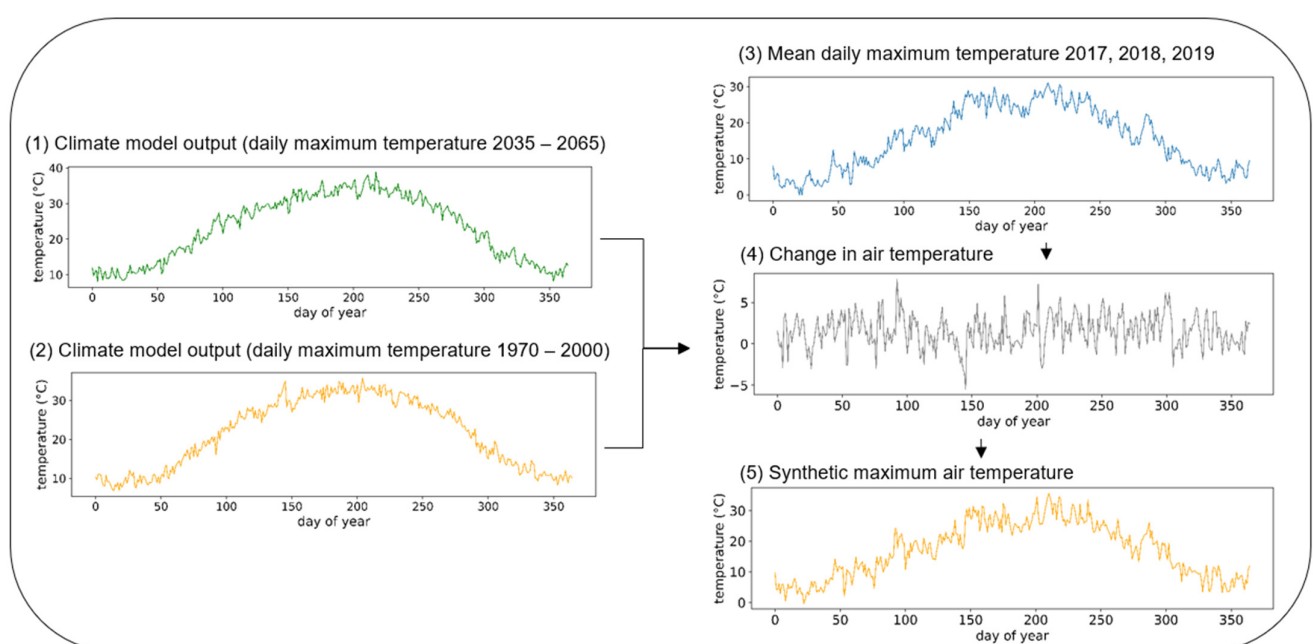

**Figure A4.** Process of generating daily synthetic temperature values based on the climate models. All line graphs represent annual air temperature curves in °C. The process comprises the derivation of daily air temperature changes (4) between historical (2) and forecasted (1) data and generation of future temperature air data (5) by calculating the sum of temperature changes (4) and averaged measured values (3).

Wind speed values are generated analogously to precipitation, but since there were very few negative values in the result (1 in RCP2.6 and RCP4.5, 2 in RCP8.5), these were set straight to zero.

Although the synthetically generated weather data for 2050 are based on the outputs of a single forecast model instead of an ensemble of different climate models, the results of the climate data preparation are consistent with the findings of regional climate models in Germany [33]. For the Berlin region, climate change leads to a general increase in maximum air temperature throughout the year with precipitation sums rising but with a trend towards more extreme events and therefore fewer wet days in summer.

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
