# Peer review of "How Would We Cycle Today If We Had the Weather of Tomorrow? An Analysis of the Impact of Climate Change on Bicycle Traffic"

_sustainability, doi:10.3390/su131810254_

Round 1
Reviewer 1 Report
Comments on "How would we cycle today if we had the weather of tomorrow?"
This manuscript tried to predict bicycle traffic in a changed climate in the city of Berlin. The climate change has been well addressed, the four time series approaches, including one traditional statistical approaches and three machine learning models, have been well introduced and applied, an excellent cross-validation procedure has been designed, the conclusion has been clearly drawn. Therefore, I would recommend it to be accepted with minor revision.
1. In Table 2, the bicycle count (sum) of 2017-2019 are statistically meaningful, partially due to the population in Berlin are rather stable (the population growth rate in 2017-2019 are 0.36%, 0.36%, and 0.13% respectively, according to https://worldpopulationreview.com/world-cities/berlin-population). To apply the approaches of the manuscript to other places (e.g., Luxembourg or Seattle, population growth rate 1.66% and 2.2% in 2020), or to longer periods (e.g., a decade, or 40 years), the population change could be a very important factor. So it is better to have some relative discussion in the text.
2. Figure 5 is very impressive, which shows every machine learning model performs better than the traditional statistical approach. Do all the numbers here were produced after fine-tuning the model? In other words, do they compare best-to-best for each model? Why SARIMAX in dashed line performs so bad? And what is the main reason for the best performance of XGBoost in solid line? Some more discussion would be better.
3. typo on Line 527 of Page 16 "weather".
Reviewer 2 Report
It was a great read I would say. The few major points I want to mention that,
1) The study title was more weather-focused, but the analysis was climate scale-focused, suggesting an update of the study title.
2) The applicability of the global climate models on a regional scale was not thoroughly checked. It needs further investigation before accepting the final findings from the prediction of the model.
3) Readers will be more interested to know about the model selection procedure, why these 4 particular models were selected, what is the rationale behind selecting these models?
4) A clear and concise discussion on the uncertainty associated with the climate data is necessary which is completely omitted.
5) Some minor comments were also provided in the attached PDF.

Round 2
Reviewer 2 Report
Impressive, the manuscript was significantly improved. I am happy to accept the manuscript in its current form.